# Bioactive Regeneration Potential of the Newly Developed Uncalcined/Unsintered Hydroxyapatite and Poly-l-Lactide-Co-Glycolide Biomaterial in Maxillofacial Reconstructive Surgery: An In Vivo Preliminary Study

**DOI:** 10.3390/ma14092461

**Published:** 2021-05-10

**Authors:** Shinji Ishizuka, Quang Ngoc Dong, Huy Xuan Ngo, Yunpeng Bai, Jingjing Sha, Erina Toda, Tatsuo Okui, Takahiro Kanno

**Affiliations:** Department of Oral and Maxillofacial Surgery, Shimane University Faculty of Medicine, Izumo, Shimane 693-8501, Japan; ishizuka@med.shimane-u.ac.jp (S.I.); dongngocquang1987@gmail.com (Q.N.D.); ngoxuanhuy158@gmail.com (H.X.N.); xyywq@126.com (Y.B.); jsswjbnjw@gmail.com (J.S.); et1211@med.shimane-u.ac.jp (E.T.); tokui@med.shimane-u.ac.jp (T.O.)

**Keywords:** bone regeneration, Runx2, osteocalcin, CD68, periostin, osteoconductivity, poly-l-lactide-co-glycolide, poly-l-lactic acid, uncalcined/unsintered hydroxyapatite

## Abstract

Uncalcined/unsintered hydroxyapatite (HA) and poly-l-lactide-co-glycolide (u-HA/PLLA/PGA) are novel bioresorbable bioactive materials with bone regeneration characteristics and have been used to treat mandibular defects in a rat model. However, the bone regenerative interaction with the periosteum, the inflammatory response, and the degradation of this material have not been examined. In this study, we used a rat mandible model to compare the above features in u-HA/PLLA/PGA and uncalcined/unsintered HA and poly-l-lactic acid (u-HA/PLLA). We divided 11 male Sprague–Dawley rats into 3- and 16-week groups. In each group, we assessed the characteristics of a u-HA/PLLA/PGA sheet covering the right mandibular angle and a u-HA/PLLA sheet covering the left mandibular angle in three rats each, and one rat was used as a sham control. The remaining three rats in the 16-week group were used for a degradation assessment and received both sheets of material as in the material assessment subgroup. At 3 and 16 weeks after surgery, the rats were sacrificed, and mandible specimens were subjected to micro-computed tomography, histological analysis, and immunohistochemical staining. The results indicated that the interaction between the periosteum and u-HA/PLLA/PGA material produced significantly more new bone regeneration with a lower inflammatory response and a faster resorption rate compared to u-HA/PLLA alone. These findings may indicate that this new biomaterial has ideal potential in treating maxillofacial defects of the midface and orbital regions.

## 1. Introduction

Bone-fixation devices are essential instruments in the daily operation of oral and maxillofacial surgical practices. Accordingly, the materials used to manufacture bone plates and screws play an important role in the development of this field as well as other skeletal surgical specialties. Titanium has long been used to stabilize bone fragments, with excellent outcomes, and has therefore become the standard bone hardware material [1]. The advantages of titanium include its strong physical characteristics and high biocompatibility. However, drawbacks include stress-shielding, thermal irritability, and infection, and these issues can persist as long as the device remains in the patient’s body [1]. Thus, the removal of titanium plates and screws is sometimes required. Developments in medical science have focused on enhancing the benefits and reducing the complications of such procedures. Therefore, although titanium is still a practical choice, several bioresorbable alternatives have been developed. These materials can be bioresorbed, precluding the need for patients to carry foreign materials in their bodies permanently.

Although the early generations of bioresorbable materials applied in maxillofacial surgery, i.e., polyglycolic acid (PGA), poly-l-lactic acid (PLLA), and their copolymers, were applied with some success, these are still less than ideal. PGA, which was the first bioresorbable material developed, has a rapid degradation rate such that the physical strength of the material is weakened before the bone-healing process is complete. In contrast, although PLLA (a representative “first-generation” bioresorbable material) has several favorable features, such as good mechanical properties and ease of processing, it also has a slow bioresorption time, which can induce a foreign-body inflammatory response [2]. The copolymer of the two materials (poly-lactide-co-glycolide or PLLA/PGA), a “second-generation” bioresorbable material, was developed to overcome these drawbacks. The resorption time of this combination can be adjusted and much reduced by modifying the PLLA:PGA ratio [3]. However, all of these three previous generations of biomaterials lack bioactivity features, such as osteoconduction and bone-bonding ability. As a result, their application in specific anatomical regions in maxillofacial reconstructive surgery, such as the paranasal sinuses or the orbit in the midfacial region, is still limited, because defects in these areas tend to reappear after the material has been resorbed. Accordingly, a new option that can overcome this issue is strongly needed.

The “third-generation” of bioresorbable materials (uncalcined and unsintered hydroxyapatite [HA]/poly-l-lactide or u-HA/PLLA) was developed in the 1990s and has been extensively examined in several animal and clinical studies. This material has good mechanical properties and is conducive to osteoconduction [4,5,6]. The bending strength and modulus of u-HA/PLLA are greater than or equal to that of human cortical bone, and the resorption time is compatible with the bone-healing process. The bioactivity of this material has also been demonstrated in various animal studies on extremity and facial bones. Moreover, numerous studies have reported an absence of postoperative wound infection and foreign body granulomas. Hence, fixation devices made of u-HA/PLLA have been used in various applications, such as the treatment of orbital wall [7,8], maxillary [1,9], and mandibular [10,11] fractures, and zygomatic complex fractures [12], as well as in orthognathic surgery [13,14]. However, Shikinami et al. [15] and Sukegawa et al. [16] reported that complete resorption of u-HA/PLLA may take up to 5.5 years. This lengthy degradation time gives rise to a higher risk of long-term adverse bodily responses, such as inflammation or implant palpation, according to studies reporting complications after implantation, especially in maxillofacial region [1,17,18]. Therefore, a shorter resorption time is necessary for maxillofacial clinical applications.

Recently, a “fourth generation” bioresorbable material has been introduced to overcome the limitations of the u-HA/PLLA composite [19]. This material, composed of polymer and bioceramic materials, was developed with the goal of maintaining the bioactive features of u-HA/PLLA while reducing the resorption time. Uncalcined and unsintered HA and poly-l-lactide-co-glycolide (u-HA/PLLA/PGA) is a combination of u-HA and the copolymer of PLLA and PGA produced by a forging process. As it is synthesized using the method described by Shikinami [4], u-HA has almost the same composition as natural bone. Ooishi et al. also reported that its particles were more bioactive and bioresorbable than those of other bioresorbable bioceramics [20]. Hence, in comparison to other HAs, such as sintered or calcined HA, u-HA has different characteristics, which makes u-HA/PLLA/PGA a novel material. According to the manufacturer, u-HA/PLLA/PGA has mechanical properties similar to those of u-HA/PLLA [4,10,11]. As the biodegradation duration of PLLA/PGA copolymer can be reduced by changing the ratio between PLLA and PGA [21], the new material is assumed to have a shorter decomposition and bioresorption period. Moreover, the presence of u-HA particles in the new composite implies that u-HA/PLLA/PGA devices will also show similar bioactive osteoconductivity. The results of our previous study showed that u-HA/PLLA/PGA and u-HA/PLLA had comparable osteoconductive abilities in a rat mandibular defect model [19], confirming the potential applicability of this material to the maxillofacial region. According to immunohistochemical (IHC) analysis of Runx2 and LepR expression, bone marrow is the main source of osteoblastic cells [19]. However, numerous investigations have indicated that periosteum, the tissue located along the outer surface of the bone, is another local source of skeletal stem cells for regenerative bone healing [22,23,24].

As u-HA/PLLA/PGA theoretically has a shorter resorption time than u-HA/PLLA, there may be an increase in the number of biodegradation products, which could induce inflammatory reactions in the host after implantation in the maxillofacial region. However, to our knowledge, there have been no previous studies on the role of periosteum in the presence of composite implants consisting of u-HA particles, the biocompatibility of the new material, or its degradation time in vivo.

To address these issues, we conducted a preliminary study with three main objectives: first, to assess periosteum-derived bone regenerative responses to u-HA/PLLA/PGA in rat mandibles via the presentation of periostin, a key extracellular matrix component of the periosteum involved in periosteum-derived bone-regenerative functions, and to compare this with the responses to u-HA/PLLA; second, to evaluate and compare the inflammatory responses to u-HA/PLLA/PGA and u-HA/PLLA in terms of CD68 expression; and third, to preliminarily measure and compare the degradation of these materials based on their molecular weight retention rates.

## 2. Materials and Methods

### 2.1. Materials

The u-HA/PLLA/PGA and u-HA/PLLA reconstructive materials were manufactured by Teijin Medical Technologies Co., Ltd. (Osaka, Japan). The forged composite sheets had the following dimensions: 5 mm length × 5 mm width × 0.3 mm thickness (Figure 1A). The u-HA/PLLA/PGA sheets consisted of u-HA (10% of total weight) and copolymer poly-l-lactide-co-glycolide (90% of total weight), and the PLLA:PGA ratio was 88:12. The u-HA/PLLA sheets consisted of u-HA (40% of total weight) and PLLA (60% of total weight). The u-HA particle size in both materials ranged from 0.2 to 20 μm (mean size, 3–5 μm). The calcium:phosphorus molar ratio was 1.69, and the CO_3_^2−^ molar level was 3.8%.

### 2.2. Surgical Procedure

Eleven male Sprague–Dawley rats (age = 10 weeks; weight = 250–270 g) were divided into two groups: material group (*n* = 9) and sham controls (*n* = 2). The material group was further divided into two subgroups: u-HA/PLLA/PGA and u-HA/PLLA, in which implantation was performed in the right and left hemi-mandibles, respectively. Hence, each subgroup had nine specimens, of which three were harvested at Week 3 and six at Week 16. At Week 16, the material sheets were detached in three specimens in each subgroup to calculate the degradation rates. In the sham control group, two specimens in one rat were harvested at Week 3, and two specimens in the other rat were harvested at Week 16 (Figure 2).

All rats underwent general anesthesia via intraperitoneal injection of medetomidine hydrochloride (0.15 mg/kg), midazolam (2 mg/kg), and butorphanol (2.5 mg/kg). All surgeries were performed under standard aseptic conditions. On each side, a 1-cm full-thickness longitudinal incision was created through the submandibular skin. The soft tissue was then dissected and retracted to expose the mandibular angle area (Figure 1B). Then, the mandibular angle was covered buccally with either a u-HA/PLLA/PGA sheet or a u-HA/PLLA sheet, as follows: each rat in the material group received one u-HA/PLLA/PGA sheet on the right side and one u-HA/PLLA sheet on the left side. The sheets were fixed in place with hemoclips, as previously described (Figure 1C) [19]. In the sham control group, the mandibular angles were not covered. The wounds were then irrigated with normal saline and closed in layers. All rats awoke 1–2 h after surgery and showed normal behavior and appetite.

At 3 and 16 weeks after surgery, the rats were euthanized via an anesthetic overdose. Each hemi-mandible was harvested and soaked in 10% neutral buffered formalin for further analysis (Figure 1D–F).

All animal experiments adhered to the Guidelines for the Care and Use of Laboratory Animals of Shimane University Faculty of Medicine, Izumo, Japan. The animal protocol was approved by the Animal Ethics Committee of Shimane University (approval number: IZ2-73).

### 2.3. Evaluation of New Bone Formation in the Buccal Side via Micro-Computed Tomography (CT) in the Material Groups

We used high-resolution micro-CT to assess the volume of new bone formation in three dimensions. Except for the six specimens used to measure the molecular weight at Week 16, the rat hemi-mandibles were scanned using a 3D Micro X-ray CT CosmoScan FX scanner (Rigaku Corporation, Tokyo, Japan) after sacrificing the animals and before the specimens were sent for hematoxylin and eosin (HE) and IHC staining. The parameters used for micro-CT were as follows: voltage, 90 kV; current, 88 μA; scan time, 2 min; field of view, 10.24 mm × 10.24 mm × 10.24 mm; matrix size, 512 × 512 × 512; resolution, 50 pixels/mm; voxel size, 20 μm × 20 μm × 20 μm; and bone mineral density phantom, 0, 50, 200, 800, and 1200 mg HA/cm^3^.

The 3D volumes of the scanned samples were generated from the acquired 2D lateral projections using Fiji software. Before analysis, scanned bone volumes were digitally reoriented using the “Transform: Rotate” plugin in Fiji (http://imagej.net/Fiji, accessed on 9 February 2021) [25]. This ensured that the axes were parallel to the plane of the material sheets (Figure 3B). After rotating the 3D volumes, it was very easy to separate the bony tissue, soft tissue, and both reconstruction material sheets by direct observation, because the level of radiopacity increased in the order of soft tissue, u-HA/PLLA/PGA sheet, u-HA/PLLA sheet, bony tissue, and titanium hemoclip.

For each specimen, we used the Fiji “ROI Manager” tool to measure the area of the outer bone in all slides in which bone was identified on the buccal side of the material (Figure 3C). Then, a new outer bone volume was calculated for each specimen using the following formula:V = **∑**Si × d
in which V is the volume (mm^3^), Si (mm^2^) is the area of new bone on the buccal side of the material in a slice, and d is the slice thickness (equal to 0.02 mm).

### 2.4. Tissue Preparation, HE Staining, and IHC Staining

#### 2.4.1. Tissue Preparation and HE Staining

The specimens collected from the material groups at Weeks 3 and 16 were decalcified, dehydrated, and embedded in paraffin. The specimens were sectioned along the coronal plane to produce each final section (Figure 1D–F) and then stained with HE for histological evaluation.

#### 2.4.2. IHC Staining of Runx2, Osteocalcin (OCN), Periostin, and CD68

The paraffin-embedded tissue specimens were sliced into 4-μm sections. The sections were deparaffinized with xylene and rehydrated with ethanol. Enzymatic antigen retrieval was performed using proteinase K (0.4 mg/mL). Then, a 3% hydrogen peroxide solution was applied to block endogenous peroxidase activity. The sections were incubated with a rabbit polyclonal anti-Runx2, anti-CD68, or anti-periostin antibody, or with a mouse monoclonal anti-OCN antibody, for 50 min at room temperature. After the sections had been rinsed three times with phosphate-buffered saline, they were incubated with Histofine Simple Stain MAX PO (MULTI) (#414191; Nichirei Biosciences Inc., Tokyo, Japan) for 30 min at room temperature. Then, the sections were incubated with diaminobenzidine (DAB) for 10 min and counterstained with hematoxylin for 30 s. All IHC analysis procedures were performed by Sept. Sapie Co., Ltd. (Tokyo, Japan). The stained slides were assessed using a BX43 light microscope (Olympus Corp., Tokyo, Japan).

### 2.5. IHC Evaluation

The expression of Runx2 and CD68 was defined using a labeling index. In each specimen, three images were taken at 40× magnification to examine the newly formed bone on the buccal side of the material sheet according to the appearance of anti-Runx2 and anti-CD68 antibodies. After storing the files in TIFF format, each image was input into the Fiji software. Then, the “Cell Counter” tool was used to record the number of positive and negative cells in the image area. The percentage of positive cells was calculated as the number of positive cells divided by the total number of cells. Finally, the labeling index for each specimen was defined as the average percentage based on three images from that specimen.

As OCN and periostin antibodies accumulate in the extracellular matrix [26], we quantified their expression in the buccal bone area (OCN) and the buccal fibrous tissue adjacent to the material sheet (periostin) using digital H-scores, as described previously [27,28,29]. Briefly, based on the principle that a higher DAB intensity reflects a higher concentration of antigen, we measured the intensity of DAB chromogen staining. Numerically, a darker DAB signal has a higher intensity and a lower value on a scale from 0 to 255. Hence, we used the digital H-score to reflect the appearance of antigens in this study. Digital H-scores were calculated using the intensity function in the Fiji software as follows:First, an empty area was selected, and its RGB values were measured. If the values were not near 255, the “Subtract Background” tool in the “Process/Subtract Background” tab was used to correct the background.Second, the area of interest was chosen using various selection tools (Figure 4A) and saved to the “ROI Manager”.Third, the “Color Deconvolution” tool in the “Image/Colour Deconvolution” tab was applied to separate the image into three panels representing hematoxylin staining (Figure 4B), DAB staining (Figure 4C), and the background, respectively.Fourth, the previously selected ROI was superimposed onto the DAB image (Figure 4D).Finally, the “Measure” tool in the “ROI Manager” tab was used to calculate DAB intensity (i), which ranged from 0 (black) to 255 (white). The digital H-score (i.e., reciprocal intensity) (f) in each specimen was then measured using the formula f = 255 − i, as described by Nguyen et al. [29].

### 2.6. Molecular Weight and Retention Rate

Molecular weight analysis was performed using an HLC-8320GPC (TOSOH Corporation, Tokyo, Japan) gel permeation chromatography instrument. Tetrahydrofuran was used as the mobile phase at a flow rate of 0.35 mL/min. The weight-average molecular weight (Mw) was determined relative to polystyrene standards (TOSOH Corporation) using refractive index detection at 40 °C.

After measuring the Mw of each material, we calculated the retention rate of each material using the equation R = Mw_16_/Mw_0_, where R is the retention rate (%), Mw_16_ is the molecular weight at Week 16 (g/mol or Da), and Mw_0_ is the initial molecular weight (g/mol or Da).

### 2.7. Statistical Analysis

Statistical analyses were performed using SPSS software for Mac OS (version 25.0; IBM Corporation, Armonk, NY, USA). We used the following two tests:The Wilcoxon signed-rank test was used to compare the volume and area of the outer bone (micro-CT and histomorphometry), labeling index (Runx2 and CD68), and digital H-scores (OCN and periostin) between the u-HA/PLLA/PGA and u-HA/PLLA subgroups at each time point.The Mann–Whitney U test was used to compare all the parameters listed above at each time point within the same subgroup.

In all analyses, *p* < 0.05 was taken to indicate statistical significance.

## 3. Results

### 3.1. Micro-CT Evaluation

In the three-dimensional assessment, due to the greater ratio of u-HA in the compound, the u-HA/PLLA sheets were detected more clearly than the u-HA/PLLA/PGA sheets. The amount of new bone formation was the same in both groups at each time point. At Week 3, newly formed bone was first seen on the buccal side of the material sheets. At Week 16, the newly formed bone nearly covered the entire outer side of the material sheets. In the sham control group, no new bone was seen at either time point (Figure 5).

The volume of new bone on the outer side of the material sheet in both groups was very low at Week 3 (0.372 and 0.168 mm^3^, respectively) and very high at Week 16 (2.895 and 3.801 mm^3^, respectively). According to the volume of outer bone, there were no significant differences in new bone formation between the u-HA/PLLA and u-HA/PLLA/PGA subgroups at each time point (*p* > 0.05). However, the mean volume of newly formed bone differed considerably between the two time points in each material group (*p* < 0.05) (Figure 6).

### 3.2. Histological Assessment

We examined the histological features of both materials at each time point. At Week 3, newly formed bone that was mostly immature bone with a low level of mineralization was seen on the buccal side of the material sheet. There was also a thin layer of fibrous tissue surrounding the material. However, the thickness of this layer was heterogeneous. For example, the area that was not in contact with the bone was thicker than the area that contacted the bone. At Week 16, bone formation on the buccal side was visible in both subgroups. The outer bone displayed characteristics of maturity that were similar to those of the host bone. However, some new immature bone was detected at the tip of the mature new bone, where contact occurred with the material sheet. At this time, the fibrous tissue surrounding the material sheet appeared to be thinner than at Week 3. However, the thickness of this layer was still heterogeneous, as observed at Week 3 (Figure 7).

### 3.3. IHC Analyses

#### 3.3.1. Runx2

The Runx2 expression patterns of the u-HA/PLLA and u-HA/PLLA/PGA subgroups were similar (Figure 8). At Week 3, Runx2-positive cells were clearly observed in the area in which newly formed bone was identified, close to the material sheets. In contrast, at Week 16, Runx2 expression was not readily observed in either material group (Figure 8).

The above observations corresponded with the results of the Runx2 labeling index. All results indicate a significant decrease in the Runx2 labeling index from Week 3 to Week 16 in both material subgroups (*p* < 0.05). However, the Runx2 labeling index did not differ between the two subgroups at any time point (*p* > 0.05) (Figure 9).

#### 3.3.2. OCN

The expression of OCN was comparable between the groups. At Week 3, we observed low expression of OCN on the outer side of the material sheets. At Week 16, we observed higher OCN expression in newly formed bone (Figure 10).

The OCN expression did not differ between the groups at any time point (*p* > 0.05). Although OCN expression in both groups was higher at Week 16 than at Week 3, this difference was not significant (*p* > 0.05) (Figure 11).

#### 3.3.3. Periostin

Periostin expression in both subgroups was visible on the buccal side of the material sheets. The distribution of periostin expression on the outer side of the material sheets was thicker and wider during the early stage than in the later stage (Figure 12).

The significantly higher digital H-scores of periostin at Week 3 versus Week 16 in each subgroup reflected the above observations (*p* < 0.05). However, the digital H-scores of periostin were similar among the subgroups at each time point (*p* > 0.05) (Figure 13).

#### 3.3.4. CD68

Cells positive for CD68 were concentrated as a layer covering the material sheets. When magnified, these layers seemed to be thinner during the later stage compared with the early stage in both subgroups. Furthermore, the layers of CD68-positive cells were positioned in the same area as the fibrous tissue identified in the histological assessment (Figure 14).

The results of the labeling index indicated no differences between the two subgroups at each time point or between the two time points within each material subgroup. However, the index was significantly lower at Week 16 than Week 3 in both subgroups (Figure 15).

### 3.4. Retention Rate

At Week 16, the average retention rate was significantly lower in the u-HA/PLLA/PGA subgroup (19.4%) than in the u-HA/PLLA subgroup (46.1%) (*p* < 0.05). This may indicate that the degradation time of the new material, u-HA/PLLA/PGA, is much shorter than that of the previous material, u-HA/PLLA (Figure 16).

## 4. Discussion

Bone-fixation devices have been constructed from various types of materials, from titanium to bioresorbable polymers. Although existing materials have been successfully implemented in several types of reconstructive surgery, their use is limited in delicate anatomical areas such as the midface or in defect-type lesions in maxillofacial reconstructive surgery [1,2]. Therefore, innovations focused on incorporating osteoconductive features into physically strong materials are necessary to improve treatment outcomes in particular situations. u-HA/PLLA is a “third generation” bioresorbable material that possesses the required physical capabilities in relation to human cortical bone and has osteoconductive bone regenerative potential, as demonstrated by in vivo studies [2]. Accordingly, it can be used in both conventional osteosynthesis surgery and to treat a defect-type lesion of the midface. However, the long resorption time of this material is still a drawback because its long-term presence inside the human body, especially in maxillofacial regions, may induce potential complications [2]. u-HA/PLLA/PGA was recently developed with the goal of shortening the resorption time with respect to its predecessor, while maintaining all of the favorable features.

As observed by micro-CT and HE staining, we detected a significant amount of new bone on the buccal side (outer side) of both tested materials. Statistical analyses revealed that the two materials did not differ in terms of the amounts of new bone created. The bone-formation mechanism of u-HA/PLLA and u-HA/PLLA/PGA can be explained by the presence of HA in the compound. A calcium phosphate layer was detected on the surface of both materials 3 days after immersion in body fluids, and this layer completely surrounded the material within 7 days [4]. This reactive layer modulated the structure and activity of the adsorbed serum protein (i.e., fibronectin). The adsorbed protein layer provided an attachment site for osteoblastic cells and their progenitors. Fibronectin binds to integrin, a cellular transmembrane protein; this interaction activates the signaling pathway, which eventually leads to osteoblastic differentiation and matrix mineralization [30]. These processes enable the two u-HA-containing materials (u-HA/PLLA and u-HA/PLLA/PGA) to bond to bone and induce new bone formation. Therefore, the u-HA component plays an essential role in the bioactive osteoconductive features of the two materials. While u-HA accounts for 40% of the composition of u-HA/PLLA, it accounts for only 10% of that of u-HA/PLLA/PGA. However, the amount of newly formed regenerative bone in the u-HA/PLLA/PGA subgroup was almost equal to or greater than that in the u-HA/PLLA subgroup, although the difference was not statistically significant (Figure 6). This phenomenon was also observed in a previous study comparing u-HA/PLLA and u-HA/PLLA/PGA in a mandibular defect model [19]. In the present study, we compared OCN expression between two materials by IHC staining. OCN is a protein secreted by osteoblasts located in the bone extracellular matrix. The expression of OCN was relatively high in, and comparable between, the two material groups from Week 3 to Week 16, with no significant differences between the two groups (Figure 11). This reflects the very early maturity of the new bone that formed on the buccal side of the materials in both groups. These characteristics were reported previously, both by our group [19,27] and others [31,32,33]. Therefore, these results indicated that the new material had comparable bioactive/osteoconductive ability to the previous material.

We used the ratio of u-HA, PLLA, and PGA described in the introduction, because the degradation time of copolymer PLLA/PGA can be modified based on the PLLA:PGA ratio [21]. For example, 50PLLA/50PGA, 75PLLA/25PGA, and 85PLLA/15PGA have resorption times of 1–2, 4–5, and 5–6 months, respectively [3]. In this study, the average retention rates of u-HA/PLLA/PGA and u-HA/PLLA confirmed that the new material has a much shorter degradation time than the previous one (Figure 16). As copolymer PLLA/PGA with a ratio of 88/12 degraded more quickly than polymer PLLA, the new material could release more u-HA particles into the surrounding tissue than the previous material. This could explain why the u-HA/PLLA/PGA sheet stimulates new bone formation like the u-HA/PLLA sheet, despite having a much smaller percentage of u-HA particles [19].

Shikinami reported that composites of u-HA particles and polymers, such as PLLA, affect both phagocytosis and hydrolysis after implantation in vivo [4]. The low proportion of u-HA in u-HA/PLLA/PGA may be beneficial for minimizing the inflammatory response because the process of u-HA bioresorption involves inflammatory cells, such as neutrophils, monocytes, multinucleated giant cells, and macrophages, as well as osteoclasts. These cells may participate in phagocytosis, which could increase the concentrations of inflammatory mediators as well as reduce the local pH, thus increasing inflammation in the host. In this study, we assessed the inflammatory responses in both test groups by monitoring CD68 expression. CD68-positive cells formed a thin layer covering both materials at both 3 and 16 weeks. The labeling indices of CD68 expression in the two groups were also stable, with a small reduction from Week 3 to Week 16. CD68 is often used as a marker of inflammation involving monocytes and macrophages, because it is expressed by macrophage lineage cells [34]. The minimal differences in CD68 labeling indices between the two groups indicated similar inflammatory responses, suggesting that the newer material (u-HA/PLLA/PGA) did not trigger adverse reactions to a greater degree than the older material. The concentration of CD68-positive cells around the materials may be explained by continuous degradation of the materials by hydrolysis. Notably, although the newer material degraded faster, as demonstrated by the lower retained molecular weight, and was therefore expected to release more u-HA particles, the inflammatory responses induced by the two materials were similar. Consequently, the low proportion of u-HA in u-HA/PLLA/PGA may ensure a lower inflammatory response while retaining the same bone regeneration capability as the other materials.

The periosteum is a major contributor to regenerative bone healing. Histologically, the periosteum consists of two layers: an outer layer composed mainly of fibroblasts and an inner cambium layer containing osteoblasts, pre-osteoblastic cells, fibroblasts, osteochondrogenic progenitor cells, and skeletal stem cells [35]. The periosteum is also well-vascularized and innervated [35]. When a stimulus is applied to the bone, the corresponding periosteum will respond with new bone formation. Periostin, a protein necessary for bone repair, may regulate the proliferation, adhesion, and differentiation of osteoblastic cells. Horiuchi et al. [36] reported that periostin is expressed mainly in the extracellular matrix of the periosteum and the periodontal ligament. Unlike other markers, periostin is specifically expressed by preosteoblasts in the periosteum [36]. In this study, we observed a high concentration of periostin in the fibrous tissue surrounding the two materials on the buccal side (where the materials were in direct contact with the periosteum) during the early stage (Week 3). Periostin expression decreased by Week 16, as shown by digital H-scores. This pattern was consistent with the discontinuation of periostin secretion during osteoblast differentiation [36]. Similarly, the expression of Runx2 in both groups decreased significantly from Week 3 to Week 6. Runx2 is a key transcription factor in osteoblast differentiation. Thus, the elevated labeling index of Runx2 at Week 3 may reflect the early stage of active osteoblast differentiation. The events related to periostin and Runx2 can be summarized as follows: after introducing the stimulus to the local area (i.e., the material was inserted during surgery), the periosteum responded by contributing osteoblastic cells and their progenitors to the affected area, leading to increased periostin expression. The osteoblastic cells interacted with the calcium phosphate layer surrounding the materials and activated the signaling pathway as discussed above. Some of the osteoblastic cells began to differentiate to prepare for new bone formation during the subsequent period, as reflected by the high expression of Runx2 during Week 3. New bone was formed around the new material, and the stimulus gradually diminished, thus reducing the number of osteoblastic cells transported from the periosteum and the corresponding osteogenic activities. As a result, the expression of periostin and Runx2 decreased considerably during the later stage (Week 16). The process demonstrated that the periosteum interacted with the two osteoconductive materials (u-HA/PLLA and u-HA/PLLA/PGA), leading to regeneration of new bone. These findings help to explain the effectiveness of u-HA/PLLA in several clinical studies [37,38,39] in which a u-HA/PLLA sheet was used to cover defects in the orbital wall. As the human orbital wall is adjacent to the midface paranasal sinuses, one side of the u-HA/PLLA sheet used to cover a defect must face the external environment. The procedural success in these studies may have depended on the formation of new bone on the inner surface of the material, which interacted with the periosteum and internal body environment. Animal studies using orbital fracture models are required to test this hypothesis.

Given the favorable features observed in this study and in previous research, the “fourth generation” u-HA/PLLA/PGA bioresorbable bioactive material may be well suited to reconstructive surgery of the midface and especially to orbital defect reconstruction, such as in orbital trauma, as well as other applications in maxillofacial reconstructive surgery. As the anatomy of this region contains many hollow structures, direct contact with the external environment is unavoidable. Although various materials have been used to correct these structures surgically, disadvantages remain. As discussed previously, the presence of titanium and non-resorbable polymer is permanent and has been associated with morbidities. Furthermore, conventional non-osteoconductive material can re-introduce a defect. Although an autologous bone graft as the gold standard is biologically ideal, this necessitates a donor site, and shaping the bone segment to match the complex three-dimensional anatomy of the orbit can be difficult. Therefore, a rapidly bioresorbable, biocompatible, osteoconductive, and easy-to-handle material, such as u-HA/PLLA/PGA, could be a preferable solution for treating defects in maxillofacial reconstructive surgery.

The most suitable bioresorbable scaffold materials for use in bone tissue engineering are synthetic biodegradable aliphatic polyesters, such as polycaprolactone (PCL) [40], polylactic acid (PLLA, PDLA, poly-d/l-lactic acid) [41,42,43], PGA [41], and their copolymer PLLA/PGA [44]. The bioactive ceramics, including HA and β-tricalcium phosphate (β-TCP), represent another group of bioresorbable scaffold materials. These ceramics are chemically similar to the native bone mineral [45,46,47]. Material design based on organic-inorganic composites not only improves the weak points of ceramic biomaterials but also provides various biological functions [48]. For example, the porous composite of uncalcined/unsintered HA and poly-d/l-lactide (u-HA/PDLLA) is a feasible biomaterial scaffold for bony regeneration because of its components including u-HA particles and polymer PDLLA. The particles of u-HA are more bioactive and more resorbable than those of other resorbable bioceramics [20]. The presence of u-HA in a composite provides the bioactivity and osteoconductive ability, which are crucial features for bone tissue regeneration [4,6,15]. Second, due to its good biocompatibility and controllable degradation rate, PDLLA is widely used in tissue engineering [49]. Hence, the composite made of 70 wt% u-HA particles and 30 wt% PDLLA (50PLLA:50PDLA) showed good bioactivity, biocompatibility, and osteoconductivity [50]. This composite could be suitable for reconstruction of bone defects caused by trauma or tumors in the maxillofacial region.

The u-HA/PLLA/PGA composite used in the present study also showed similar features to the 3D u-HA/PDLLA composite. Due to the presence of two enantiomeric isomers of polylactic acid, PLLA/PGA is only one of the copolymers of PLA and PGA. In a review, Gentile et al. [44] summarized the physicochemical properties and field of application of different PLLA/PGA and PDLA/PGA copolymers. Although these copolymers have lower mechanical strength than PLLA polymer, they have faster degradation rates. For example, PLLA has a degradation time of 12–18 months, tensile modulus of 2.7 GPa, and crystallinity of 37% [44], whereas Jose reported that 85PLLA/15PGA had a degradation time of 5–6 months and tensile modulus of 2.0 GPa and exist as an amorphous polymer [51]. However, when dispersing u-HA particles into PLLA/PGA, the u-HA/PLLA/PGA in the present study also showed the same mechanical strength as u-HA/PLLA, which was shown to be superior to PLLA polymer [4]. On the other hand, although PLLA/PGA does not possess bioactivity or osteoconductivity, its composite with u-HA particles showed comparable bone regeneration capacity to u-HA/PLLA [19]. Moreover, the organic components of PLLA and PGA are easily hydrolyzed and have low immunogenicity and low toxicity in the human body [52]. Therefore, u-HA/PLLA/PGA can be considered as a promising bioresorbable material to design porous scaffolds due to its biocompatibility, osteoconductivity, and biodegradability. Due to its superior mechanical properties and biological features, this composite can be fabricated as a thin mesh/sheet, which can effectively repair bone defects in extremely thin bony walls, such as the orbital or sinus wall. In addition, because the mechanical properties and degradation time could be regulated by changing the ratio of components in the composite, u-HA/PLLA/PGA could have more medical applications.

A limitation of this new material is its limited radio-opacity. As shown in Figure 5A, the visibility of u-HA/PLLA/PGA is quite poor due to the low proportion of u-HA in the overall composition (10%). This characteristic may increase the difficulty of clinical re-evaluation of the material using conventional X-ray technology.

This study had several limitations. First, we used a rat mandible model, and the results achieved may not be similar to those in humans. Second, the sample size was small, which may have reduced the statistical power. Finally, the molecular weight data were available only for the 16-week samples, so the preceding and consequent degradation processes were not assessed. Further studies with larger sample sizes and more assessment time points are needed to further examine the characteristics of this newly developed biomaterial.

## 5. Conclusions

The results of the present study indicated that the regenerative bone interaction between the periosteum and the new u-HA/PLLA/PGA material is beneficial for maxillofacial reconstruction, with a significant amount of bioactive osteoconductive new bone regeneration. u-HA/PLLA/PGA shows great potential as a rapidly bioresorbable material with high biocompatibility and a low inflammatory response. These features may render this new biomaterial an ideal choice for reconstructive surgery of the midfacial region.

## Figures and Tables

**Figure 1 materials-14-02461-f001:**
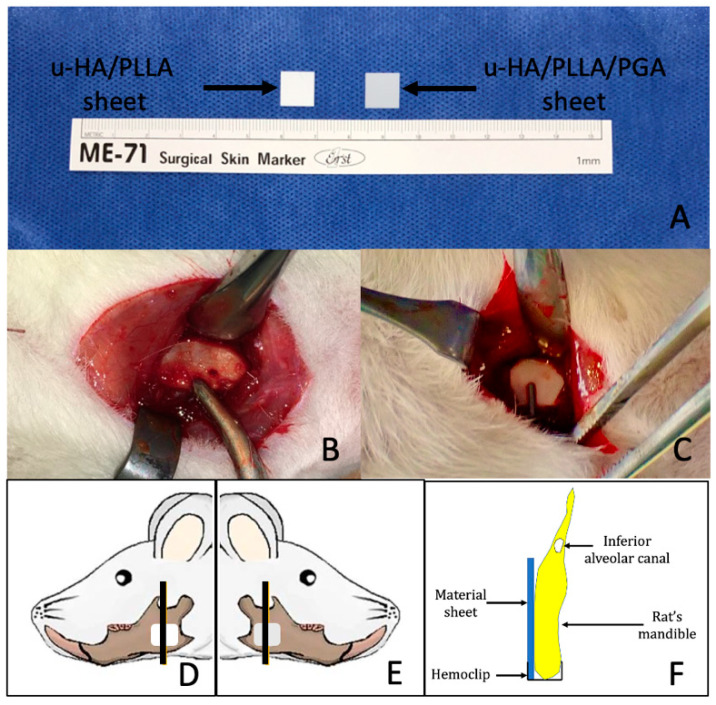
Surgical procedure. (**A**) u-HA/PLLA sheet (left) and u-HA/PLLA/PGA sheet (right) with the same dimensions: 5 mm length × 5 mm width × 0.3 mm thickness. (**B**) The mandibular angle area was exposed on the right side. (**C**) Fixation of the reconstruction material using a hemoclip. The black vertical lines in (**D**,**E**) indicate the sites at which samples were collected for analysis. u-HA/PLLA material was applied on the left side and u-HA/PLLA/PGA material on the right side. (**F**) Schematic coronal view of one specimen.

**Figure 2 materials-14-02461-f002:**
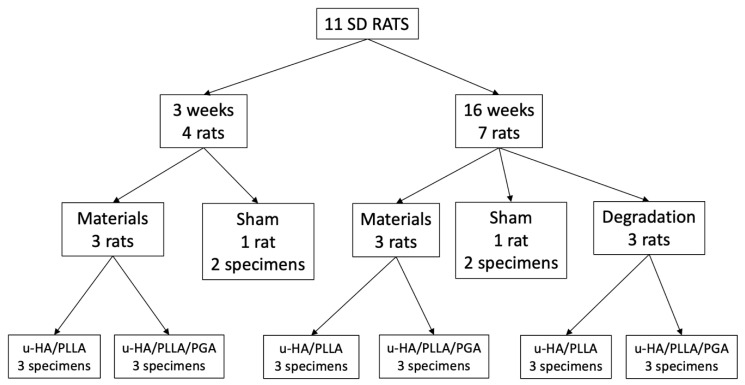
Experimental groups.

**Figure 3 materials-14-02461-f003:**
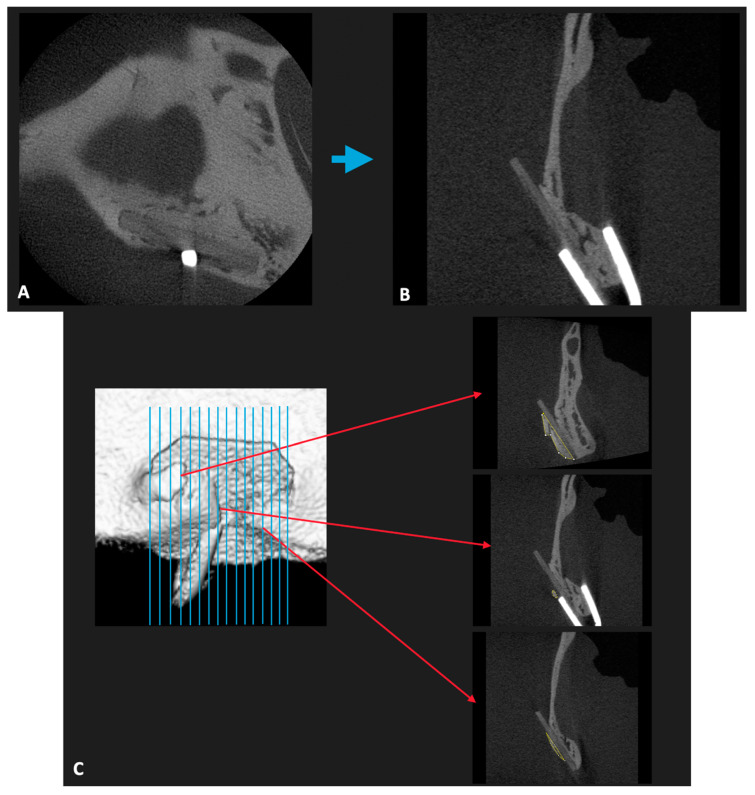
Micro-CT assessment. (**A**) Image of the initial micro-CT data. (**B**) Illustration of a slice image in which the outer bone was easily detected after reorienting. (**C**) Illustrated overview of one specimen. The area of newly formed bone on the buccal side of the material sheet was measured in each slice using the Fiji “ROI Manager” tool.

**Figure 4 materials-14-02461-f004:**
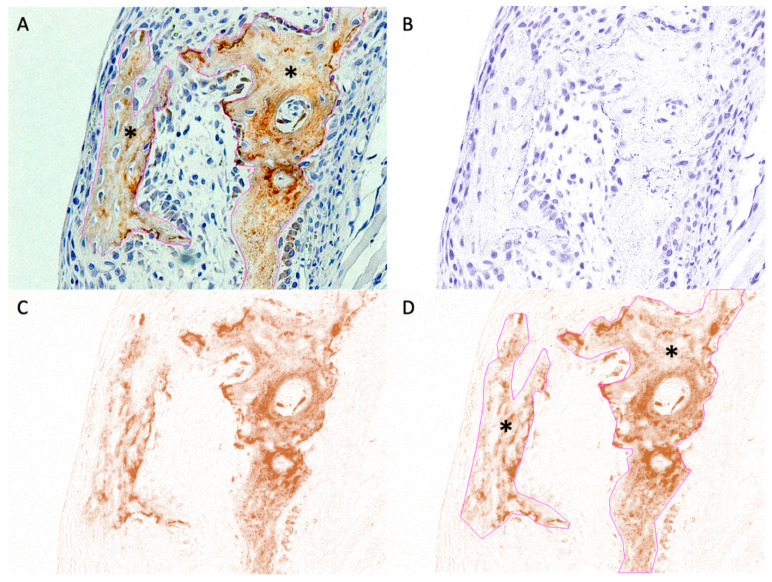
Example of ROI selection for areas containing new bone (anti-OCN IHC staining). (**A**) Original image with overlay of the selected ROI. *, selected area. (**B**) Hematoxylin-stained image separated from the original image. (**C**) DAB-stained image separated from the original image. (**D**) Superimposition of the saved ROI onto the DAB-stained image. *, selected area.

**Figure 5 materials-14-02461-f005:**
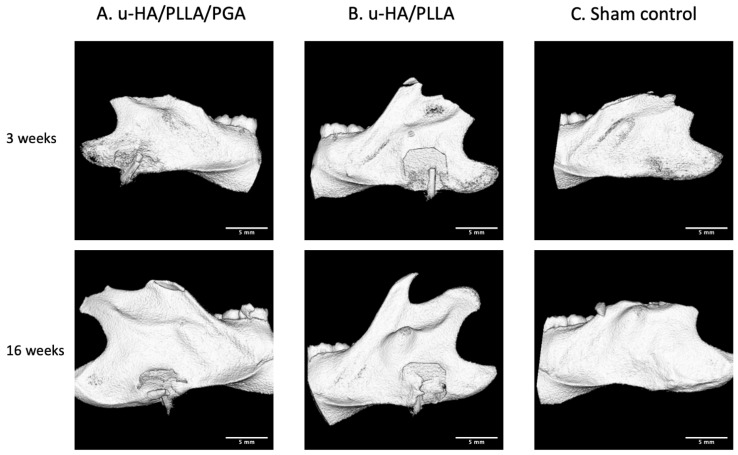
Micro-CT images of rat hemi-mandibles. The u-HA/PLLA/PGA is not visible due to the low proportion of u-HA. 3D view of the (**A**) u-HA/PLLA/PGA subgroup, (**B**) u-HA/PLLA subgroup, and (**C**) sham control group. Scale bar: 5 mm.

**Figure 6 materials-14-02461-f006:**
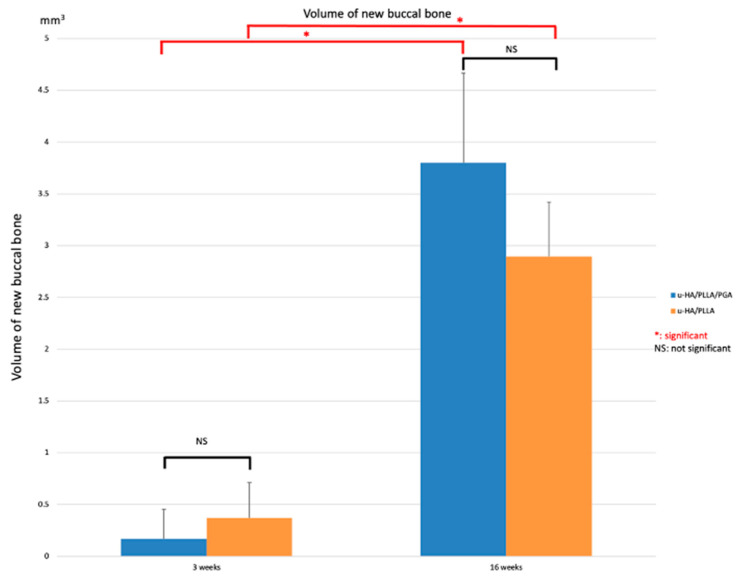
Volume of new bone, as determined by micro-CT evaluation, in the u-HA/PLLA/PGA and u-HA/PLLA subgroups. * *p* < 0.05.

**Figure 7 materials-14-02461-f007:**
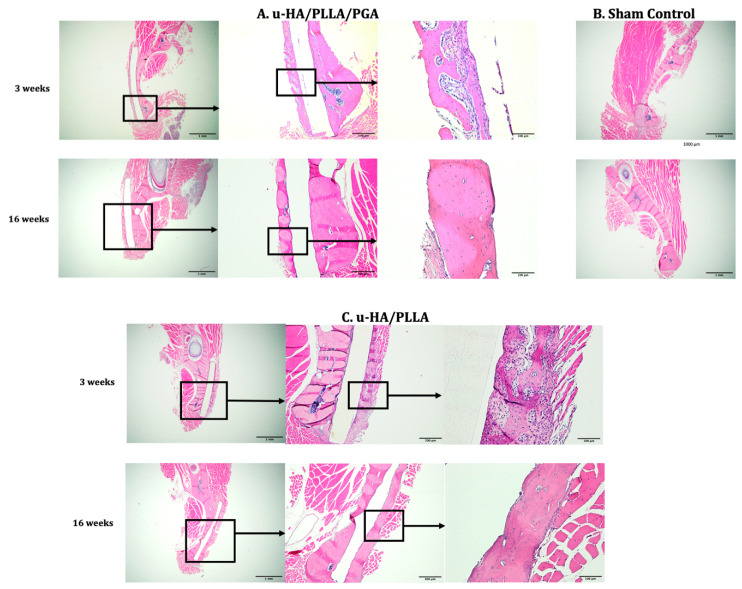
Hematoxylin-and-eosin-stained sections from the u-HA/PLLA/PGA subgroup, u-HA/PLLA subgroup, and sham control. Images of each subgroup were taken at 1.25×, 4×, and 20× magnification (from left to right). Images of the sham control group were taken at 1.25× magnification. (**A**) u-HA/PLLA/PGA subgroup. (**B**) Sham control group. (**C**) u-HA/PLLA subgroup. Scale bar: 1 mm (1.25× magnification), 500 μm (4× magnification), and 100 μm (20× magnification).

**Figure 8 materials-14-02461-f008:**
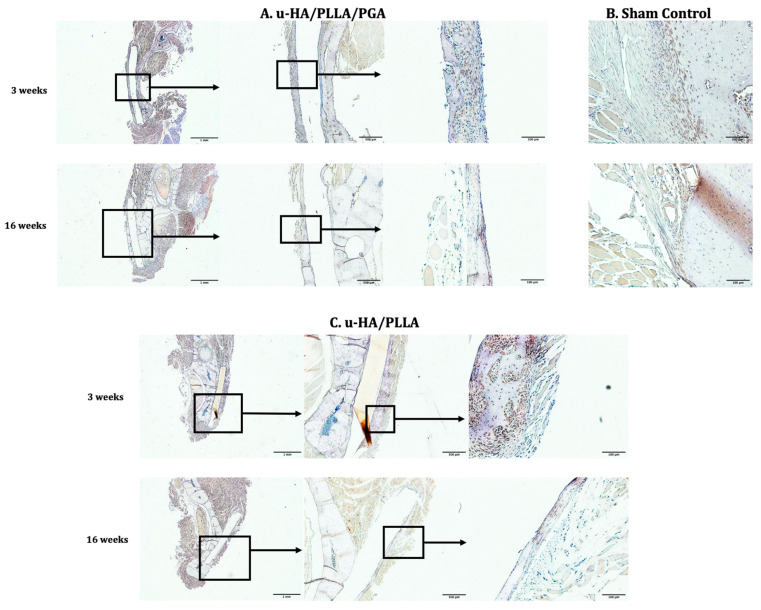
Runx2 expression in the u-HA/PLLA/PGA subgroup, u-HA/PLLA subgroup, and sham control. Images in each subgroup were taken at 1.25×, 4×, and 20× magnification (from left to right). Images of the sham control group were taken at 20× magnification. (**A**) u-HA/PLLA/PGA subgroup. (**B**) Sham control group. (**C**) u-HA/PLLA subgroup. Scale bar: 1 mm (1.25× magnification), 500 μm (4× magnification), and 100 μm (20× magnification).

**Figure 9 materials-14-02461-f009:**
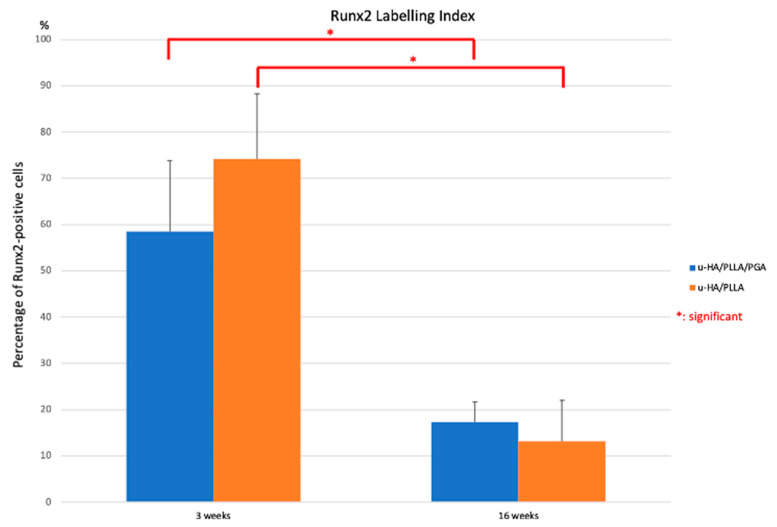
Runx2 labeling index in the u-HA/PLLA/PGA and u-HA/PLLA subgroups. * *p* < 0.05.

**Figure 10 materials-14-02461-f010:**
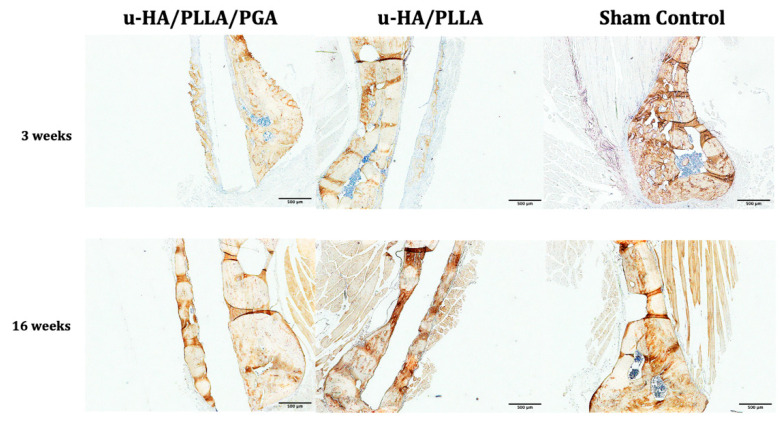
OCN expression in the u-HA/PLLA/PGA subgroup, u-HA/PLLA subgroup, and sham control. All images were taken at 4× magnification. Scale bar: 500 μm.

**Figure 11 materials-14-02461-f011:**
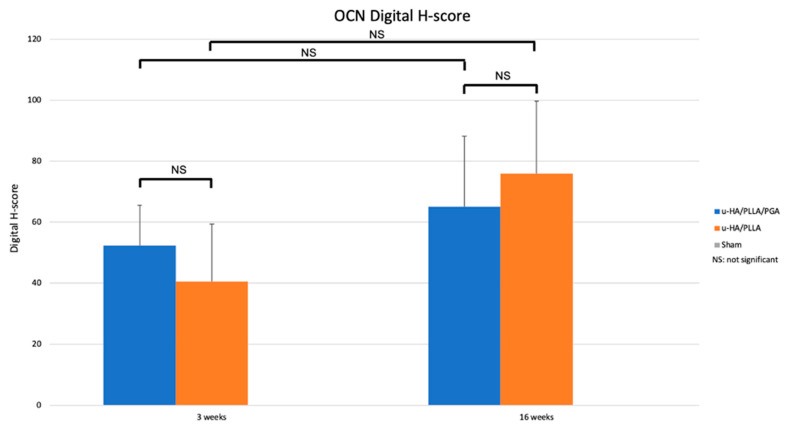
Digital H-scores based on IHC staining using an anti-OCN antibody in the u-HA/PLLA/PGA and u-HA/PLLA subgroups.

**Figure 12 materials-14-02461-f012:**
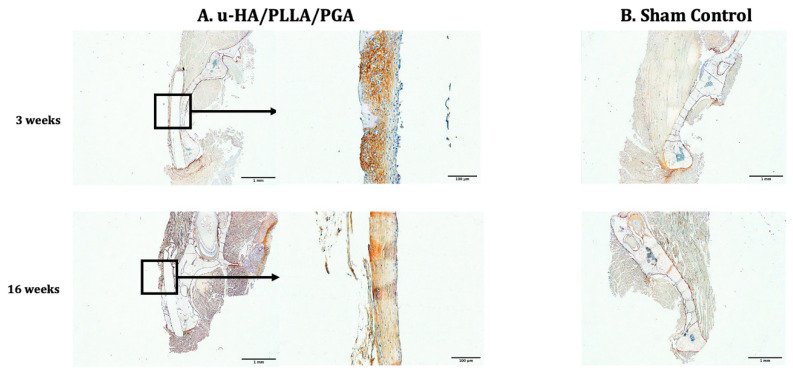
Periostin expression in the u-HA/PLLA/PGA subgroup, u-HA/PLLA subgroup, and sham control. Images in each subgroup were taken at 1.25× and 20× magnification (from left to right). Images of the sham control group were taken at 1.25× magnification. (**A**) u-HA/PLLA/PGA subgroup. (**B**) Sham control group. (**C**) u-HA/PLLA subgroup. Scale bar: 1 mm (1.25× magnification) and 100 μm (20× magnification).

**Figure 13 materials-14-02461-f013:**
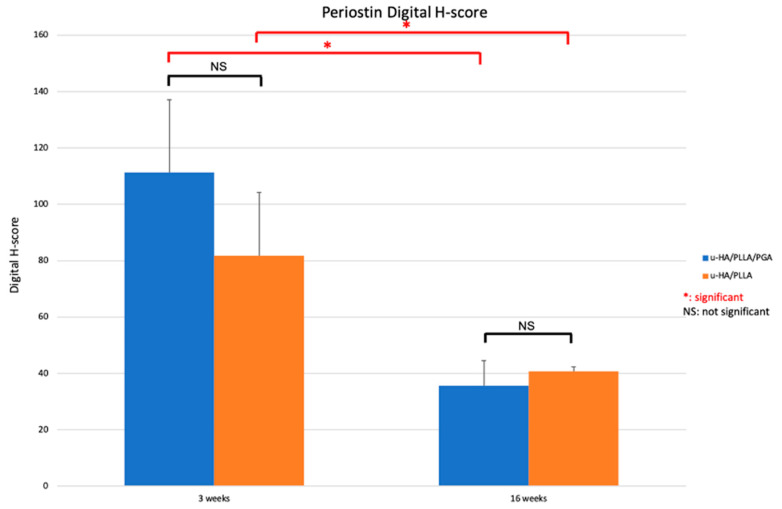
Digital H-scores based on IHC staining using an anti-periostin antibody in the u-HA/PLLA/PGA and u-HA/PLLA subgroups. * *p* < 0.05.

**Figure 14 materials-14-02461-f014:**
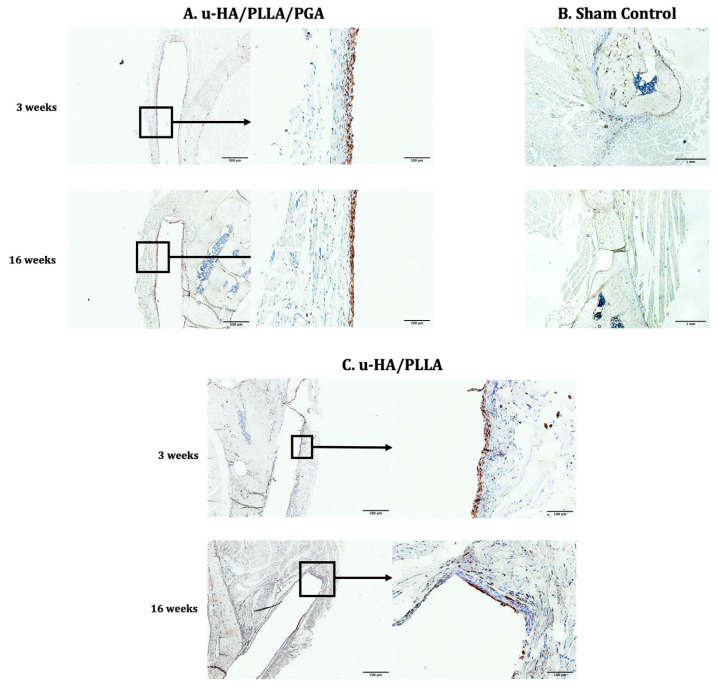
CD68 expression in the u-HA/PLLA/PGA subgroup, u-HA/PLLA subgroup, and sham control. Images in each subgroup were taken at 4× and 20× magnification (from left to right). Images of the sham control group were taken at 1.25× magnification. (**A**) u-HA/PLLA/PGA subgroup. (**B**) Sham control group. (**C**) u-HA/PLLA subgroup. Scale bar: 1 mm (1.25× magnification image), 500 μm (4× magnification image), and 100 μm (20× magnification image).

**Figure 15 materials-14-02461-f015:**
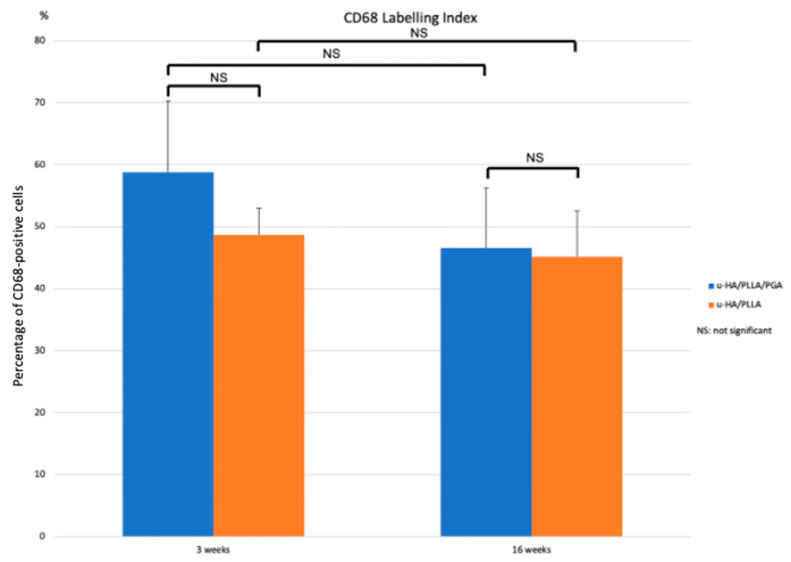
Labeling index based on IHC staining using an anti-CD68 antibody in the u-HA/PLLA/PGA and u-HA/PLLA subgroups.

**Figure 16 materials-14-02461-f016:**
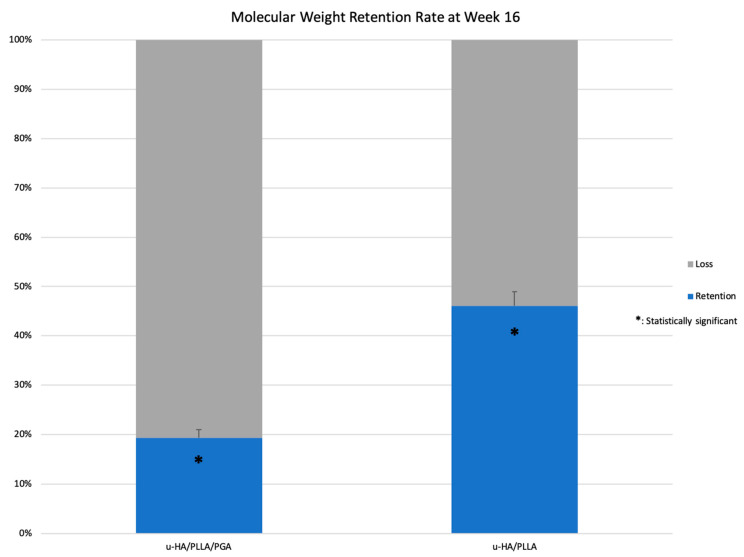
Retention rates of u-HA/PLLA/PGA and u-HA/PLLA. * *p* < 0.05.

## Data Availability

All data have been illustrated in the manuscript.

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
