# Peer review of "Bioactive Regeneration Potential of the Newly Developed Uncalcined/Unsintered Hydroxyapatite and Poly-l-Lactide-Co-Glycolide Biomaterial in Maxillofacial Reconstructive Surgery: An In Vivo Preliminary Study"

_materials, 2021, doi:10.3390/ma14092461_

Round 1

Reviewer 1 Report

The manuscript has been well written and results are presented well. However, some minor issues/suggestions remain which need to be addressed:

  • Please include the details of parameters used for microCT scanning.
  • For microCT analyses, authors need to describe how the proper range of thresholding was selected to separate newly formed bone from other tissues/implants?
  • It would be good to discuss the potential applications of HA/PLLA/PGA to fabricate porous bone scaffolds.
  • Lines 113-115: The authors need to clarify why these specific ratios are selected for u-HA, PLLA, and PGA?
  • Figures 6, 9, 11, and 13: The vertical axis does not have any label.

Author Response

RESPONSE TO COMMENTS/SUGGESTIONS FROM REVIEWERS

Dear Reviewer 1

Thank you very much for your insightful comments and suggestions. We here incorporated all of these from you into our revised manuscript and would like to answer your comments in the following paragraphs.

Sincerely yours,

The corresponding author,

Prof. Takahiro Kanno, DDS, FIBCSOMS-ONC/RECON, PhD

Comments:

The manuscript has been well written and results are presented well. However, some minor issues/suggestions remain which need to be addressed:

  • Please include the details of parameters used for microCT scanning.
  • For microCT analyses, authors need to describe how the proper range of thresholding was selected to separate newly formed bone from other tissues/implants?
  • It would be good to discuss the potential applications of HA/PLLA/PGA to fabricate porous bone scaffolds.
  • Lines 113-115: The authors need to clarify why these specific ratios are selected for u-HA, PLLA, and PGA?
  • Figures 6, 9, 11, and 13: The vertical axis does not have any label.

Answer:

Thank you very much for your kind and constructive comments regarding our manuscript. We would like to answer your comments as follows:

  • About the details of parameters used for microCT scanning: We agree with your comment. The details of parameters used for micro CT scanning have been described in the section from Line 161 to Line 164.
  • About selecting a proper range of thresholding to separate newly formed bone from other tissues/implants: We are very pleased with your comment. In fact, when using Fiji software to assess the volume data of specimens, we utilized the function of Threshold Tool to calculate the threshold of soft tissue, u-HA/PLLA/PGA sheet, u-HA/PLLA sheet, boney tissue, and titanium hemoclip. The results of thresholds for them are ranging from (-294; 1258), (296; 1693), (1258 to 3747), (1258 to 3747), and (3747; +∞), respectively. Because there are overlapped ranges between boney tissue and reconstruction material sheets in the image, we can not select the proper range of threshold to separate the newly formed bone from reconstruction material sheets. After 3 weeks, the newly formed bone turned to mature, so the density of the newly formed bone was similar to that of the parent bone. Only depending on the threshold is impossible to distinguish these objects. Fortunately, we noticed that by direct observation, we can separate the boney tissue, soft tissue, and both reconstruction material sheets. Hence, in our study, we did not mention the selecting range of threshold to calculate the newly formed bone volume. With this issue, we just added the section from Line 168 to Line 171. In the next study, we will use the better method to evaluate the structures in micro-CT.
  • About the potential applications of HA/PLLA/PGA to fabricate porous bone scaffolds: Thank you for your exact We think your idea regarding the potential applications of this material for manufacturing porous bone scaffolds could be very valuable for bone regeneration. We fully agree with your suggestion. By comparing the similar features with u-HA/PLLA and u-HA/PDLLA, the composites which have been proven as a feasible scaffold material to design the porous scaffold, we noticed that u-HA/PLLA/PGA could be considered as a promising bioresorbable bioactive material to fabricate the porous scaffold due to its biocompatibility, osteoconduction, and biodegradation ability. This discussion has been added in the section from Line 482 to Line 519.
  • About the specific ratios of u-HA, PLLA, and PGA: We think your comment relating to the specific ratios of each component in the new materials is very insightful. We have been discussed this issue in the section from Line 403 to Line 432. Your sincere suggestions helped us to improve the content of our articles.
  • About the label of the vertical axis in Figures 6, 9, 11, and 13: Thank you for your suggestion. We corrected Figures 6, 9, 11, 13, and 16 to make those charts more standard

Once again, thank you for your kind and detailed comments.

Reviewer 2 Report

The Review numbered 1198121 with title “Bioactive Regeneration Potential of the Newly Developed Uncalcined/Unsintered Hydroxyapatite and Poly-L-Lactide-Co-Glycolide Biomaterial in Maxillofacial Reconstructive Surgery: An In Vivo Preliminary Study” is a very interesting study.

I suggest to the authors some minor revisions before pubblication.

  • The results should be better clarified and discussed.
  • The materials and methods should be better described.

Author Response

RESPONSE TO COMMENTS/SUGGESTIONS FROM REVIEWERS

Dear Reviewer 2

Thank you very much for your kind comments. We would like to answer your comments in the following paragraphs.

Sincerely yours,

The corresponding author,

Prof. Takahiro Kanno, DDS, FIBCSOMS-ONC/RECON, PhD

Comments:

The Review numbered 1198121 with title “Bioactive Regeneration Potential of the Newly Developed Uncalcined/Unsintered Hydroxyapatite and Poly-L-Lactide-Co-Glycolide Biomaterial in Maxillofacial Reconstructive Surgery: An In Vivo Preliminary Study” is a very interesting study.

I suggest to the authors some minor revisions before pubblication.

  • The results should be better clarified and discussed.
  • The materials and methods should be better described.

Answer:

Thank you very much for your kind comments regarding our manuscript. We would like to revise the manuscript as follows:

  • About the results should be better clarified and discussed: Thank you for your suggestion. We have made them more clearly in the discussion section from Line 395 to Line 432.
  • About materials and methods should be better described: We totally agree with your comment. We have added the sections from Line 161 to Line 184 and from Line 168 to Line 171.

Once again, thank you for your kind suggestions.

Detailed manuscript revision:

  • Modifying the content of the information section from Line 77 to Line 78 and from Line 52 to Line 104 to make this section more clear about the purposes of this study.
  • Adding the details of the parameters used for micro CT scanning from Line 161 to Line 164.
  • Adding the section from Line 168 to Line 171 to explain the method used to assess the volume of the new bone.
  • Correcting the Figures 6, 9, 11, 13, and 16.
  • Adding the section from Line 395 to Line 432 to discuss the specific ratios of each component of the new composite material and its advantages.
  • Adding the section from Line 482 to Line 519 to discuss the potential applications in bone tissue engineering of u-HA/PLLA/PGA to fabricate the porous bone scaffold.

Reviewer 3 Report

In the manuscript entitled “Bioactive regeneration potential of the newly developed uncalcined/unsintered hydroxyapatite and poly-L-lactide-co-glycolide biomaterial in maxillofacial reconstructive surgery: an in vivo preliminary study” authors described preliminary study on using uncalcined/unsintered hydroxyapatite and poly-L-lactide-co-glycolide biomaterial in maxillofacial reconstructive surgery. The work includes research carried out on live animals. Presented results demonstrate that the bone regenerative interaction between the periosteum and u-HA/PLLA/PGA material plays a role for maxillofacial reconstruction.

The manuscript is undoubtedly interesting, however some deficiencies exist, which are listed below:

The authors should give a more clear and updated introduction focusing on the main objective of this study. In the first two paragraphs, the authors spent quite a lot of words introducing that in recent decades, polymers have received great interest and have been widely explored. That is the general knowledge especially of bone tissue engineering, but it is not directly relevant to the present study, where the material is hybrid organic-inorganic HAp/PLLA/PGA. So it should be given in a more concise way. The authors should give a more convincing introduction why the research is meaningful and important. For example, the authors should refer to previous studies and talk about what has been done and what have not been achieved.

In the discussion I am missing reference and discussion of major work on mechanical, structural and biological properties of composites for bone tissue engineering, no reference to this substantial contribution to this area is given but furthermore, that research should be discussed in context to yours as it is highly relevant, for example ACS Appl. Mater. Interfaces 2011, 3, 1692–1701; Micron 2018, 119, 64–71;  but many others are available.

I could not find any information on the u-HA/PLLA/PGA characteristics in the manuscript, even if it is commercial material, it requires at least minimal analysis or reference to the manufacturer's reports. At the moment, it is not known what the authors really studied. The definition of PLLA is very broad, as it is known, the microstructure of the polymer has a significant impact on the properties of the biomaterials. Moreover, it is not clear what kind of phase/phases uHAp contains.

Author Response

RESPONSE TO COMMENTS/SUGGESTIONS FROM REVIEWERS

Dear Reviewer3

Thank you very much for your insightful comments and suggestions. We here incorporated all of these from you into your revised manuscript and would like to answer your comments in the following paragraphs.

Sincerely yours,

The corresponding author,

Prof. Takahiro Kanno, DDS, FIBCSOMS-ONC/RECON, PhD

Comments:

In the manuscript entitled “Bioactive regeneration potential of the newly developed uncalcined/unsintered hydroxyapatite and poly-L-lactide-co-glycolide biomaterial in maxillofacial reconstructive surgery: an in vivo preliminary study” authors described preliminary study on using uncalcined/unsintered hydroxyapatite and poly-L-lactide-co-glycolide biomaterial in maxillofacial reconstructive surgery. The work includes research carried out on live animals. Presented results demonstrate that the bone regenerative interaction between the periosteum and u-HA/PLLA/PGA material plays a role for maxillofacial reconstruction.

The manuscript is undoubtedly interesting, however some deficiencies exist, which are listed below:

  • The authors should give a more clear and updated introduction focusing on the main objective of this study. In the first two paragraphs, the authors spent quite a lot of words introducing that in recent decades, polymers have received great interest and have been widely explored. That is the general knowledge especially of bone tissue engineering, but it is not directly relevant to the present study, where the material is hybrid organic-inorganic HAp/PLLA/PGA. So it should be given in a more concise way. The authors should give a more convincing introduction why the research is meaningful and important. For example, the authors should refer to previous studies and talk about what has been done and what have not been achieved.
  • In the discussion I am missing reference and discussion of major work on mechanical, structural and biological properties of composites for bone tissue engineering, no reference to this substantial contribution to this area is given but furthermore, that research should be discussed in context to yours as it is highly relevant, for example ACS Appl. Mater. Interfaces 2011, 3, 1692–1701; Micron 2018, 119, 64–71;  but many others are available.
  • I could not find any information on the u-HA/PLLA/PGA characteristics in the manuscript, even if it is commercial material, it requires at least minimal analysis or reference to the manufacturer's reports. At the moment, it is not known what the authors really studied. The definition of PLLA is very broad, as it is known, the microstructure of the polymer has a significant impact on the properties of the biomaterials. Moreover, it is not clear what kind of phase/phases uHAp contains.

Answer:

Thank you very much for your kind and constructive comments regarding our manuscript. We would like to answer your comments as follows:

  • About the introduction section: We are very pleased with your comment. In fact, this material is developed to improve the drawback of u-HA/PLLA, a bioresorbable material used to fabricate successfully the rigid bone fixation devices. Hence, in the first two paragraphs, we wanted to introduce the historical background of different rigid bone fixation systems and emphasize that this material is not only used to stabilize bone fracture segments but also accelerate bone healing. Because u-HA/PLLA/PGA is a novel material, there is a lack of study about it. This is the reason why we conducted this preliminary study to verify its biocompatibility, bioactive osteoconductivity, and biodegradation rate compared with u-HA/PLLA material. So, we have modified the introduction in the section from Line 77 to Line 78 and from Line 82 to Line 104 to make the introduction section better. On the other hand, we all agree with you about the potential applications of this novel material for bone tissue engineering. This section will be present in the discussion section.
  • About the discussion of potential applications of u-HA/PLLA/PGA for bone tissue engineering: Thank you for your exact comment. We think your idea regarding the potential applications of this material for bone tissue engineering is very valuable. We fully agree with your suggestion. Because this novel material has feasible features such as biocompatibility, osteoconduction, biodegradation ability, we noticed that u-HA/PLLA/PGA could be considered to be a promising bioresorbable material to design the porous scaffold. Your comment suggested we discuss further potential applications of u-HA/PLLA/PGA and gave us some relevant valuable references as Reference No.47 and 48. This discussion has been added in the section from Line 482 to Line 519.
  • About the information on u-HA/PLLA/PGA characteristics: We think your comment relating to the information on u-HA/PLLA/PGA characteristics is very insightful. So, we have analyzed the characteristics of this material in the discussion section from Line 395 to Line 432.

Once again, thank you for your kind and detailed comments.

Round 2

Reviewer 3 Report

The authors improved their manuscript significantly, it is now suitable for publication and I recommend publishing this manuscript.